# Adiponectin Role in Neurodegenerative Diseases: Focus on Nutrition Review

**DOI:** 10.3390/ijms21239255

**Published:** 2020-12-04

**Authors:** Rita Polito, Irene Di Meo, Michelangela Barbieri, Aurora Daniele, Giuseppe Paolisso, Maria Rosaria Rizzo

**Affiliations:** 1Department of Advanced Medical and Surgical Sciences, University of Campania “Luigi Vanvitelli”, Piazza Miraglia 2, 80138 Naples, Italy; rita.polito@unicampania.it (R.P.); irenedimeo@libero.it (I.D.M.); michelangela.barbieri@unicampania.it (M.B.); giuseppe.paolisso@unicampania.it (G.P.); 2Department of Environmental Biological Pharmaceutical Sciences and Technologies, University of Campania “Luigi Vanvitelli”, Via G. Vivaldi 42, 81100 Caserta, Italy; aurora.daniele@unicampania.it; 3CEINGE-Advanced Biotechnologies Scarl, Via G. Salvatore 486, 80145 Naples, Italy

**Keywords:** Adiponectin, neurodegenerative diseases, Alzheimer’s disease, Parkinson’s disease, gut microbiota, gut-brain axis, healthy nutrition

## Abstract

Adiponectin is an adipokine produced by adipose tissue. It has numerous beneficial effects. In particular, it improves metabolic effects and glucose homeostasis, lipid profile, and is involved in the regulation of cytokine profile and immune cell production, having anti-inflammatory and immune-regulatory effects. Adiponectin’s role is already known in immune diseases and also in neurodegenerative diseases. Neurodegenerative diseases, such as Alzheimer’s disease and Parkinson’s disease, are a set of diseases of the central nervous system, characterized by a chronic and selective process of neuron cell death, which occurs mainly in relation to oxidative stress and neuroinflammation. Lifestyle is able to influence the development of these diseases. In particular, unhealthy nutrition on gut microbiota, influences its composition and predisposition to develop many diseases such as neurodegenerative diseases, given the importance of the “gut-brain” axis. There is a strong interplay between Adiponectin, gut microbiota, and brain-gut axis. For these reasons, a healthy diet composed of healthy nutrients such as probiotics, prebiotics, polyphenols, can prevent many metabolic and inflammatory diseases such as neurodegenerative diseases and obesity. The special Adiponectin role should be taken into account also, in order to be able to use this component as a therapeutic molecule.

## 1. Introduction

Brain aging is typical in the elderly patient, thus representing a probable continuum with neurodegenerative diseases. In addition to brain aging, other factors, such as genetic and environmental factors, influence neurodegenerative diseases. Consequently, neurodegenerative diseases are regarded as a manifestation of accelerated aging. Conversely, aging is a major risk factor for neurodegeneration. The most known neurodegenerative diseases, Alzheimer’s disease (AD) and Parkinson’s disease (PD), are prevalent in the elderly. Various factors such as lifestyle and other environmental factors influence these diseases. Among these, obesity is a strong predisposition factor for neurodegenerative diseases. In particular, the adipose tissue through the production of adipokines, such as Adiponectin, has an active role in the development of many diseases and also in neurodegenerative diseases. Adiponectin has numerous beneficial effects, increasing anti-inflammatory mediators’ production and it also has neuroprotective effects.

Another important aspect in the development of neurodegenerative diseases is the role of gut microbiota. Nutrition and healthy lifestyle are able to act on gut microbiota populations with beneficial effects. Many data from the literature report the characterization of commensal and pathogenic bacteria in equilibrium, which is strongly influenced by diet and lifestyle having a strong impact on human health [1]. In fact, the type of diet, through the digestion of food, fermentation, metabolism, and the introduction of vitamins with the diet, impact and influence the composition of our microbiota with beneficial and/or harmful effects on our health [2]. Furthermore, it is known that the alteration in the composition of the intestinal microbiota often correlates with various symptoms, such as anxiety, depression, pain syndrome, or correlates with some brain diseases, such as neurodegenerative disorders, cognitive decline or cerebral vascular diseases [1,3]. In addition, the gut microbiota modulates the development and homeostasis of the central nervous system in the context of the immune, circulatory, and neuronal pathways [4]. Thus, it is increasingly evident that dynamic changes in the gut microbiota can alter the physiology and behavior of the brain. Therefore, the purpose of this review is first of all to clarify the biological interaction between the gut, the microbiota and the central nervous system, formerly known as the “gut-microbiota-brain”axis, also emphasizing the role of adiponectin towards this mechanism and therefore focusing on how nutrition can influence the state of health.

## 2. Adiponectin

Adipose tissue is no longer considered a storage tissue today but is considered an endocrine organ, capable of secreting a large number of factors that positively or negatively regulate many physiological functions. Adiponectin is one of the products secreted mainly by adipose tissue. Adiponectin, post-translationally modified from a 30 kDa monomeric protein into several multimers (low molecular weight or trimer, medium molecular weight or hexamer, and high molecular weight), is subsequently secreted into the circulation. By binding to its receptors, AdipoR1, AdipoR2 and T-cadherin, this adipokine activates a series of signal transduction events, including the phosphorylation of adenosine monophosphate (AMPK) and p38 mitogen-activated protein kinase (p38 MAPK), and increases the activity of peroxisome proliferator-activated receptor alpha (PPARα) [5].

The main metabolic functions of Adiponectin are exerted on the adipose tissue, liver, and muscle, affecting the glucose and fatty acids metabolism [6,7]. Indeed, Adiponectin increases glucose uptake by adipocytes and myocytes, insulin sensitivity, free fatty acids oxidation in muscle, while reducing hepatic neo-glucogenesis and preventing the increase in free fatty acids and triglycerides [8,9] (Figure 1). Studies in vitro and in vivo have highlighted the anti-inflammatory and anti-atherogenic effects of adiponectin, linked to its action both suppressing the production of pro-inflammatory cytokines and modulating the expression of the anti-inflammatory cytokine IL-10 on various cell types such as monocytes and macrophages [9,10,11]. On the other hand, TNF-α and other inflammation markers, such as IL-6, C-reactive protein, and glucocorticoids, affect adiponectin levels [9,10,11]. The anti-atherogenic effects include the modulation of the inflammatory response of the endothelium and a direct effect on the antagonist by inhibition of monocyte adhesion, of macrophage transformation, and cell proliferation on smooth muscles [12].

Another important role of Adiponectin is immunity. Indeed, it is considered as an immune mediator. Adiponectin is strongly involved in innate and adaptative immune response. Given the presence of adiponectin receptors (AdipoRs) on the surface of many immune cells, Adiponectin is able to modulate its function and response [13]. When it binds to AdipoR1 and AdipoR2, Adiponectin modulates the proliferation and polarization of immune cells. In fact, on the one hand, AdipoR1 activation modulates the suppression of proliferation and polarization of M1 macrophages and the expression of pro-inflammatory cytokines, while AdipoR2 activation modulates the polarization of anti-inflammatory M2 macrophages. Instead, T-cadherin is essential for stimulatory effects on proliferation [8]. The data literature reported that there is a strong link between adiponectin and immune cells: adiponectin increases cold-induced browning on subcutaneous adipose tissue through the proliferation of M2 macrophages by activating Akt, resulting in the activation of beige cells [9].

In addition to the regulation of immune cells, adiponectin plays a critical role in the regulation of energy expenditure and insulin sensitivity through innate mechanisms dependent on the immune response [9].

In particular, the metabolic, immune and anti-inflammatory properties of adiponectin are evident in obesity. Adiponectin concentrations are reduced in the serum of obese subjects. These lower serum levels of adiponectin in obese patients are related to several metabolic changes, such as glucose uptake, lipid metabolism and insulin sensitivity [9].

More studies have shown a different role of adiponectin in immune and autoimmune diseases. In the literature, it is reported that adiponectin is closely associated with immune diseases, both immunodeficiency and autoimmune diseases [11,12,13]. Published data reported that serum adiponectin levels increase in autoimmune diseases while decreasing in immunodeficiency diseases [9,11,12,13].

The causes of this different modulation of Adiponectin in immunodeficiencies and autoimmune diseases and the different anti or pro-inflammatory actions of Adiponectin are not yet totally clear [8]; the explanation is that, in addition to the different types of immune cells and mechanisms that intervene in immunodeficiencies compared to the autoimmune diseases, different Adiponectin isoforms intervene [7,11,12,13,14]. Indeed, the low molecular weight adiponectin oligomers seem to induce pro-inflammatory actions of adiponectin in autoimmune diseases, while high molecular weight adiponectin oligomers induce anti-inflammatory actions of adiponectin, especially in immunodeficiencies [15]. In effect, it seems that low molecular weight isoforms mainly carry out pro-inflammatory actions whereas the ones with high molecular weight possess anti-inflammatory actions [16]. In diseases characterized by immunodeficiency, lower serum levels of Adiponectin negatively correlate with the severity of the disease. Pecoraro et al. showed that patients affected by common variable immunodeficiency (CVID) have Adiponectin serum levels lower when compared to control subjects [12,13]. Moreover, they showed that Adiponectin receptor expression is differently modulated in B lymphocytes, monocytes, and NK cells of CVID patients [11]. On the contrary, in autoimmune disorders, a strong up-regulation of Adiponectin has been largely demonstrated. Indeed, in rheumatoid arthritis, high serum levels of Adiponectin are expressed in huge quantities and related to the disease progression [12,13,14]; it seems that Adiponectin exerts pro-inflammatory actions, inducing pro-inflammatory cytokines production, and activating the pathway of Nf-kβ protein complex [17,18]. Although the Adiponectin action on the immune system is largely known, molecular pathway mechanisms are not yet clear. Given the strong involvement of Adiponectin on different immune cells and then in immune response, the alteration of adipocytes results in an alteration of Adiponectin production (Figure 1).

## 3. Neurodegenerative Diseases

Neurodegenerative diseases are characterized by a chronic and selective process of cell death in neurons, which occurs mainly in relation to oxidative stress and neuroinflammation [19]. The exact etiology underlying the pathogenic process of these diseases is not yet clear. In particular, these diseases are characterized by extreme variability but, nevertheless, neuroinflammation and increased intestinal permeability are common features [20]. Serres et al. showed that inflammatory factors, such as TNF-alpha, iNOS, and IL-6, have a role in the neurodegeneration pathogenesis of the central nervous system [21].

Parkinson’s and Alzheimer’s disease are the neurodegenerative diseases most represented. Parkinson’s disease (PD) is a progressive neurodegenerative disorder affecting older adults. The causes of PD are not fully known. Genetic causes are present in a small percentage of the population (5–10% of the PD population), while monogenetic causes are rare (leucine-rich repeat kinase LRRK2 mutations). Therefore, the main feature is a degeneration of dopaminergic neurons in the substantia nigra, which results in the loss of dopaminergic function. The reduction in this function causes both motor and non-motor symptoms. Indeed, these patients, at rest, have tremors, rigidity, and bradykinesia, also associated with depression and anxiety, cognitive impairment, and autonomic dysfunction. Among the various symptoms, prodrome symptoms, such as constipation, are important [20,21,22,23,24,25].

About 50% of PD patients show delayed gastric emptying, constipation, and, in addition, other bowel-related comorbidities including Crohn’s disease and inflammatory bowel syndrome. Conversely, reduced bowel movements are related to a 2.7-fold risk increase in PD.

Moreover, the intestinal lumen can contribute to the depletion of dopamine [16]. In fact, normally, the enteric nervous system produces dopamine, but when PD occurs, the intestinal pathology observed in most cases of PD becomes an important risk factor that can exacerbate dopamine depletion and worsen PD. Cases of PD show altered intestinal homeostasis, including increased oxidative stress, leading to loss of bowel function and systemic inflammation.

Thus, in PD patients, especially during early stages in the course of the disease, the gut microbiota appears modified and characterized by an abundance of numerous bacterial groups, including Enterococcaceae, Bacteroidetes, Clostridium spp, Lactobacillaceae, Faecalibacterium prausnitzii, Prevotella and a reduction in Bifidobacterium and Bacteroides fragilis [23,24,25,26].

Moreover, in PD patients, alterations in the gut microbiota are significantly associated with worsening of symptoms associated with PD [26]. However, the molecular mechanisms causing an alteration in gut microbiota are unclear.

In addition, recent studies show that neurotrophin levels are reduced. Among these, brain-derived neurotrophic factor (BDNF) plays a central role in brain neuronal mechanisms and differentiation of midbrain dopaminergic neurons [27], influencing mood and cognitive functions. Furthermore, BDNF also regulates the immune system [28,29,30]. BDNF is also released in the GI tract, from enterocytes, enteric glial cells, and neurons, representing a neurotransmitter/neuromodulator in enteric neuronal circuitries, thus generating a control of peristalsis and gut motility. In effect, an adequate level of BDNF is essential for the expression and proper functioning of the N-methylD-aspartate receptor (NMDA) in the central and enteric nervous systems. Studies conducted in post-mortem brain tissue of PD patients showed a significant reduction, through kynurenine pathway [31], in BDNF and a reduction in the level of the membrane-bound NMDA receptor, thus contributing to both the intestinal alteration and constipation in PD patients [31].

Alzheimer’s disease (AD), the most common form of dementia in the elderly, is a chronic and irreversible neurodegenerative disease. AD patients have severe brain dysfunction as a change in learning and memory, behavioral problems, all leading to severe disability in daily activities [32,33]. AD pathogenesis is widely believed to be characterized by neuronal loss and progressive impairment of synaptic function, but mostly, and by the production and deposition of the β-amyloid peptide (Aβ) outside or around neurons, associated with an accumulation of hyperphosphorylated protein tau within cortical. This scenario favors microtubule destabilization, synaptic deficiency, disruption of Ca2 and homeostasis in neurons, and, ultimately, neuronal apoptosis [34]. Unfortunately, the mechanisms behind AD are unclear and current therapies improve symptoms only modestly [34]. Some authors believe that AD pathogenesis is associated with a peripheral infectious origin, similar to herpes simplex virus type 1 (HSV1) infection (in mice), causing neuroinflammation [34,35]. In addition, the infections with spirochete, fungi, and chlamydia pneumonia are considered potentially responsible for AD [36,37]. Likewise, recent studies have implicated gut microbiota in the etiology of AD [34,35,36,37]. Indeed, as reported by Miklossy, the infectious agents can initiate the degenerative process in AD, sustain chronic inflammation, and lead to progressive neuronal damage and amyloid deposition. The accumulated knowledge, views and hypotheses proposed to explain the pathogenesis of AD fit well with an infectious origin of the disease. The outcome of infection is determined by the genetic predisposition of the patient, by the virulence and biology of the infecting agent, and by various environmental factors, such as exercise, stress and nutrition [38]. Furthermore, there is a significant association between Alzheimer’s disease (AD) and various types of spirochete and other pathogens such as Chlamydophila pneumoniae and herpes simplex virus type-1 (HSV-1). Exposure of mammalian neuronal and glial cells and organotypic cultures to spirochetes reproduces the biological and pathological hallmarks of AD [38]. Recently, Kobayashi and colleagues demonstrated that an improvement in cognitive impairment with a simultaneous reduction in amyloidosis occurs when oral administration of Bifidobacterium short-strain A1 in a mouse model of AD is performed [39,40]. In humans, Vogt and collaborators demonstrated an association between the appearance of AD and an altered microbiota [41]. So, these studies underline that adequate microbiome can contribute to AD improvement.

In the post-mortem brain tissue of AD patients, as well as in vitro AD models, reduction in BDNF has also been shown [42]. Due to recent evidence showing that BDNF levels are regulated by the gut microbiota, it would be important to understand whether the altered microbiota can also contribute to a reduction in BDNF in AD and thereby exacerbate the neuropathology of AD, oxidative stress, and alter intestinal homeostasis in AD, as indeed happens in the PD. Indeed, as shown in a murine model by Chen et al., BDNF regulates colonic mucosal barrier function and affects gut microbiota where the expression of ZO-1, occludin, claudin-1 and claudin-2 plays an important role in maintaining the intestinal mucosal function [43].

Although AD begins as early as 20 years before the symptoms manifest [44], these murine and human studies suggest that the AD course can be affected through microbiota modification, which, in turn, is also important in AD prevention, especially for those who have a genetic predisposition.

## 4. The Gut–Microbiota–Brain Axis

Recently, a specific role of the human digestive system in brain function has been proposed. The relationship between microbiota and its brain interaction is the so-called “gut-brain” axis. In particular, the “gut-microbiota-brain” axis indicates the biochemical signaling between the enteric nervous system (ENS) of the gastrointestinal tract and the central nervous system (CNS) [45]. The vagus nerve and/or chemical mediators released in the periphery, through a direct and indirect mechanism, respectively, represent the action pathways of the “gut-microbiota-brain” axis [46,47,48,49].

The vagus nerve (VN) is a component of the parasympathetic nervous system that actively participates in homeostasis and interactions between gut microbiota and brain. It is known that nerve disturbances are the evidence of central nervous system dysfunction, such as mood disturbances or neurodegenerative diseases, but also gastrointestinal disorders, such as irritable bowel syndrome [46,47,48,49,50]. Previous studies have indicated that vagal efferent fibers regulate responses of the gastrointestinal system through the release of neurotransmitters [50]. Less activation of the vagal nerve causes excessive production and activation of neurotransmitters, thus compromising the digestive process and affecting gastric motility [47,48,49]. Furthermore, immune effects regulated by the vagal nerve were also observed. Studies have shown that M1 macrophage activation and increased proinflammatory cytokine levels induced by abdominal surgery are relieved by vagal electrical stimulation [50,51].

One of the actions of gut microbiota is the immune signaling involvement, including the host’s neuro-immune status. In fact, imbalance of the gut microbiota and/or a dysbiosis (increase in non-commensal microbes) determines a homeostasis alteration and disruption of the gut microbiota, also altering the signaling between the gut–brain axis and producing neurological and neuroimmune alterations. The dysbiosis consequences are the compromise of the intestinal epithelium and an increased systemic inflammation with consequent up-regulation of IL-1, IL-6, and TNF-a [52]. When a systemic inflammatory state is established, the blood–brain barrier (BBB) is altered, favoring consequently a state of neuroinflammation. Biesmans et al. demonstrated that a single intraperitoneal injection of TNF-α in mice increases serum and brain levels of pro-inflammatory cytokines (TNF-α, IL-6 and MCP-1) [53].

Therefore, BBB plays a particular role. BBB is a selectively permeable membrane that regulates the passage of various molecules into the microenvironment of the neurons, through specialized endothelial cells and with the aid of multiple cellular transport channels present along the membrane, thus separating the central nervous system from the peripheral blood [54,55]. BBB endothelial cells’ structural disruption can expose the CNS to harmful sub-stances in circulation, contributing to CNS diseases. Many studies highlighted some patients’ psychiatric disorders, such as anxiety, depression, autism disorders, PD, AD and schizophrenia, to be related to microbe-induced BBB dysfunction [56,57]. Unfortunately, the microbiota mechanism influencing the physiology of BBB remains unknown. A potential mechanism influencing the BBB is the production, by gut microbiota, of metabolites and neurotransmitters that can alter CNS function. In addition, decreased gut microbiota or a dysbiosis can result in increased permeability of the BBB in rodent models [58,59].

Luczynski et al. demonstrated that microbiota composition is involved in brain morphology, in particular in hippocampal and microglial, in a germ-free mouse model [60]. Bao et al., instead, showed that alteration in gut microbiota and gut mucosa occurring in Crohn’s disease correlates negatively with cortical thickness, as evidenced by using magnetic resonance [61]. In other studies, the authors highlighted the active role of gut microbiota in the production of amines and neuroactive amines [62,63,64,65].

Moreover, the gut produces most of the serotonin thus influencing various systems (digestive, nervous, immune), and heart function, while gut microbiota by stimulating enterochromaffin cells of the colon, contributes to the 5- -hydroxytryptamine (5-HT)level in the colon and blood [22,66,67,68]. In recent studies, the authors showed that 5-HT levels and dopamine levels increased after administering Lactobacillus in germ-free mice, suggesting the possibility of using bacterial transplants to treat diseases such as PD [69] and confirming that dopamine synthesis can be modulated via the microbiota–gut–brain axis [70,71]. Although most biogenic amines are produced by bacterial–host interaction, a group of intestinal bacteria produces these molecules directly. Pessione et al. showed that Lactobacillus spp. and Enterococcus produce and release histamine and tyramine in the intestinal lumen, while Escherichia coli and Pseudomonas produce endogenous GABA [72].

As shown in previous studies, in order to achieve immune competence and inflammatory state of gastrointestinal mucosal, microbiota plays a critical role.

In fact, commensal bacteria preserve both gastrointestinal health and correct development of mucosal immunity, unlike those germ-free conditions in which develop deficits of gut-associated lymphoid tissues (GALT). Through the production of anti-inflammatory cytokines, such as IL-10 and IL-13, Bacteroides fragilis and other members of the Clostridia genus are able to activate an anti-inflammatory state, differently from what occurs if pathogenic bacteria, such as Salmonella typhimurium and Clostridium difficile, produce cytokine pro-inflammatory [73]. Due to structural constituents of bacterial cell walls, on the surface of the intestinal mucosa, results evermore activated a “continued functioning basal state” of the innate immune system [74], which contrariwise becomes compromised if dysbiosis and inflammation state appear [75]. This last condition can favor the appearance and/or worsening of diseases, including neurocognitive dysfunction, sleep abnormalities, and chronic fatigue [76].

In addition to innate immune response activation, the gut microbiota can directly activate cell-mediated immunity and, indirectly, mucosal immunity. In support of this, more studies confirm that T-Helper cells, TH1 and TH17 cells, are found in the small intestine [77,78,79]. Furthermore, a dysbiosis can also indirectly affect mucosal immunity by negatively dysregulating energy homeostasis and promoting oxidative stress and mitochondrial dysfunction [80]. A protective effect in inflammatory reactions occurs when gut microbiota produces butyrate and propionic acid, with increased T-reg by modifying the Foxp3 promoter [81], thus contributing to the beneficial effects in neuroinflammation [82,83]. Lastly, it is important to point out that genetics and dietary factors and antibiotic use influence the composition of the gut microbiota [84,85] (Figure 2).

## 5. Adiponectin Role in Neurodegenerative Diseases

Adiponectin secreted by adipose cells has numerous systemic beneficial effects. It is known that this adipokine is able to act on metabolic pathways, increasing insulin sensitivity, and regulating glucose levels and fatty acid reduction [86]. Furthermore, it acts as an inflammatory and immune mediator, regulating inflammatory cytokines production, and immune cell proliferation [9]. In addition, Adiponectin has anti-atherogenic properties while its reduction and/or the presence of gene polymorphisms represent risk factors for coronary heart disease and cardiovascular disease [87]. The Adiponectin role in neurological and cognitive disorders is unclear. Nevertheless, the presence of Adipo receptors in the brain suggests a pathway of Adiponectin signaling related to neurologic function [88]. Scientific data show the involvement of Adiponectin, both in cerebrovascular disorders and/or neurodegenerative diseases, such as PD and AD, as well as in chronic inflammatory diseases such as multiple sclerosis [14]. In addition, data literature reported that the Adiponectin in cerebrospinal fluid inversely correlates to AD progression. Furthermore, Adiponectin was also colocalized with p-tau in neurofibrillary tangles in AD, suggesting that sequestering may occur, which could explain the reduced CSF adiponectin levels [89,90].

In particular, a reduction in Adiponectin serum levels correlates inversely with the severity of the neurodegenerative diseases. Generally, lower serum Adiponectin levels are associated with worse cognitive function. De Franciscis et al. showed a significant positive association between the highest serum Adiponectin levels and better cognitive function during the postmenopausal time. The same highest serum Adiponectin levels resulted in the main determinant of cognitive attentional capacity in postmenopausal females, representing when it is low, an early serum marker of cognitive decline [91].

Furthermore, it is known that Adiponectin is involved in various physiological functions of the brain, such as energy homeostasis, neuronal excitability, and synaptic plasticity, aimed at reducing both amyloid-β (Aβ) and promoting neuroprotection [92].

In AD, the Adiponectin role is poorly understood, because scientific data showed its decrease or increase, or no significant changes [93]. However, most studies showed the presence of AdipoR1 in the hypothalamus and in the Meynert basal nucleus, suggesting a feasible Adiponectin involvement in the brain function pathways [94], mainly with neuroprotective properties [94]. In particular, the AD pathogenesis is characterized by the brain AMPK pathway deregulated, which in turn could phosphorylate Tau protein, causing changes in brain functioning. Through AMPK, Adiponectin can increase neuronal insulin sensitivity, increasing p-Akt through AdipoR1. Conversely, the chronic Adiponectin deficit inactivates AMPK, reducing neuronal insulin sensitivity, and inducing AD in elderly mice [95]. Insulin resistance represents a basal mechanism in diabetes pathogenesis, such as oxidative stress and inflammation. All these variables play an important role also in the pathogenesis of AD. Indeed, an increase in Aβ production and tau phosphorylation is related to glycogen synthase 3 (GSK3) activity induced in turn by insulin resistance and hyperinsulinemia. The same hyperinsulinemia reduces the clearance of Aβ in the brain with a competitive receptor mechanism [96].

Although in aging, Adiponectin serum levels reduction was associated with neurodegeneration [97], in AD patients increasing in Adiponectin serum levels could suggest a compensatory mechanism against neurodegeneration [98]. In PD patients, the role of Adiponectin is controversial. Katakoa et al. reported that Adiponectin is likely to play roles in the composition of lipid rafts, acting often as a differentiating marker between PD and alpha-synucleinopathy from PSP [99].

In conclusion, the whole role of Adiponectin in neurodegenerative disease remains unknown. Anyhow, Adiponectin has an important role, not only for its systemic properties but above all for its role in the brain, demonstrated both by the presence of receptors in the brain and by the alterations of serum levels in numerous neurodegenerative diseases.

## 6. The Interplay between Adiponectin and Gut Microbiota

The development of the adipose tissue is influenced by lifestyle and physical activity, even if in the last few years the role of gut bacteria is becoming more and more described, and already by itself is able to communicate with different organs. In fact, the gut microbiota regulates metabolic pathways, and also Adiponectin profile [100,101], inflammatory status, and then the development of obesity, type 2 diabetes, liver disease, cancer, and even neurological disorders [102].

The microbiota and any changes in the composition of the gut microbiota correlate with the dysfunction of the gut barrier, inducing an altered host immune response, low-grade inflammation [103], and changes in bile acids, gut peptides, short-chain fatty acids (SCFA) and branched-chain amino acids [104].

So, gut microbiota regulates metabolic function [105]. Membrez et al. reported that in mice, antibiotic treatment correlated with a lower fat mass and higher circulating levels of Adiponectin [106]. Decreased Adiponectin levels are instead associated with obesity, diabetes, and cardiovascular diseases [107,108]. In obesity, besides altered adipokine secretion profile, high levels of adipokines pro-inflammatory are found. Kloting et al. hypothesized that, in obesity, gut microbiota modulation may represent a strategy to modulate the adipokine profile [109]. In fact, in obese mice, the use of antibiotics, and the consequent modification of the gut microbiota, improved the levels of adiponectin and resistin, through the expression of related genes, producing a reduction in body weight even with a diet high in content fats [110].

Thus, gut microbiota and/or a possible gut microbiota dysbiosis contribute to the regulation of the secretion of adipokines, in particular, leptin and adiponectin. In white adipocytes of mice, Tryptophan-derived metabolites produced by the gut microbiota control the miR-181 family expression in order to regulate energy expenditure, insulin sensitivity, and Adiponectin secretion. The gut microbiota–miR-181 axis dysregulation represents a central mechanism on which diet and environmental changes act [111].

Additionally, gut microbiota and Adiponectin levels are influenced by gender. Indeed, some authors found a higher Firmicutes/Bacteroidetes ratio, an increase in Lachnospira and Roseburia, and higher GLP-1 plasma levels in pre-menopausal women than in postmenopausal women [112]. In contrast, they observed a lower presence of the Prevotella, Parabacteroides and Bilophila genera, and IL-6, MCP-1, and Adiponectin plasma levels in pre-menopausal women than in postmenopausal women [113,114].

In conclusion, nutrition, physical activity, healthy lifestyle influence both gut microbiota and Adiponectin expression. In particular, new strategies in nutritional and non-nutritional approach, targeting the gut microbiota, and adipokines levels, such as Adiponectin, could represent a new potential therapy of prevention of not only metabolic disorders but also immune-inflammatory disorders such as neurodegenerative diseases (Figure 3).

## 7. Nutrition and New Therapeutic Opportunities

The gut microbiota is an essential system to maintain physiological processes and health status [4,20]. For these reasons, an alteration in gut microbiota is responsible for the development of metabolic and inflammatory diseases such as type 2 diabetes, obesity, and neurodegenerative diseases. Furthermore, gut microbiota is influenced by a healthy lifestyle. In addition, the optimal functions of the nervous system require a healthy and balanced diet capable of providing a constant supply of macronutrients and micronutrients, so the possible prevention of many neurological diseases is based, above all, on a healthy diet. It is known, in fact, that obesity and inadequate eating habits have negative implications on the composition of the microbiota, on health, on cognitive development, and on neurodegeneration [115,116]. Diets rich in fruit, whole grains, vegetables and fish have been shown to be beneficial for brain function by reducing intestinal inflammation and neurodegeneration. In particular, the Mediterranean-type diet schemes have shown neuroprotective effects [117,118]. Many data from the literature, in fact, report that in patients suffering from Alzheimer’s and/or atherosclerosis, the Mediterranean diet has significant beneficial effects [119].

Indeed, inadequate eating lifestyles, such as a diet high in fat and carbohydrates, increase the risk of brain dysfunction [120]. For example, the excessive amount of fat is associated with an increase in Firmicutes and Proteobacteria and a reduction in Batteroidetes [121], also inducing an increase in plasma and fecal levels of acetate which, in turn, stimulate insulin secretion and ghrelin, causing a desire for additional food intake. Conversely, polyphenols found in fruit and the increase in Akkermansia muciniphila reduce the obesogenic effects and inflammation caused by high levels of fat.

Thus, the gut microbiota is influenced by dietary composition, so that switching from a high-fat or high-sugar dietary model to a low-fat or high-fiber dietary model can positively modify the microbiome [122]. A diet consisting mainly of animal proteins and saturated fats is associated with an abundance of Bacteroides, while a diet consisting of carbohydrates and sugars is associated with Prevotella [123]. Plant-based diets may increase SCFAs, causing an increase in Prevotella and some fiber-degrading Firmicutes [124,125].

The brain can also be functionally influenced by dietary patterns and lifestyle habits and in particular by food-derived metabolites. Just the latter, fermented by microorganisms, are released from the gut microbiota into the blood by interacting with the host, contributing to the onset of various disorders, including brain diseases.

Moreover, the host-microbiota can be modified by prebiotics, a class of non-viable substrates that serve as nutrients for the host microorganisms. The prebiotic is more specific, is not extensively metabolized, but activates a metabolism influencing microorganisms within the ecosystem [126]. Many fermentable carbohydrates have a prebiotic effect, but the most widely documented dietary prebiotics, which are beneficial to human health, are non-digestible oligosaccharides, especially fructo-oligosaccharides (known as FOS) and among them, inulin is the prebiotic of greatest interest. Other substances such as galacto-oligosaccharides also fall into the category of prebiotics. These are preferably used by bifidobacteria [127,128]. Indeed, bifidobacteria through some enzymes, the β-fructanosidase and β-galactosidase, activate the digestion of fructan and galactan oligosaccharide bonds. Food prebiotics are sometimes naturally present in foods, particularly in breast milk, or are added to them for their organoleptic properties (modification of the structure) or to obtain functional indications. In this regard, it is interesting to note that the first oligosaccharides that have positive prebiotic effects on gastrointestinal, metabolic, and immunological systems are present in breast milk, also preventing and protecting the gut epithelium from infectious pathogens [129,130]. Ultimately, the goal is to use prebiotics as an approach to improve health and reduce the risk of disease, starting already from the first years of life. Instead, the probiotics are able to modify gut microbiota. They are products by live or attenuated microorganisms, such as bacteria or yeast, and they are commercially available diversifying from each other according to bacterial composition, and biological activity. The probiotics have many health benefits, such as competitive inhibition of non-commensal bacterial growth, improving immunity, and the production of many of the essential vitamins and cofactors needed for human health [131,132,133].

Natural probiotics also include yogurt and some fermented foods. An anxiety-like mood was detected in mice with low BDNF levels infected with the nematode parasite Trichuris muris, but after administration of the probiotic Bifidobacterium longum, the anxious state disappeared and BDNF levels normalized [134].

In any case, probiotics affect the homeostasis of the central nervous system [132,133,134]. It is very likely that the neurochemical and behavioral effects are mediated by the vagus nerve, as a communication pathway used by bacteria in the activation of communication between the intestine and the brain.

Critical for homeostasis of the gastrointestinal tract are short-chain fatty acids [135]. The three main SCFAs produced by gut microbiota are acetate, propionate, and butyrate implicated in proper gut function and in protection against colitis and colorectal cancer [136]. Consequently, variations in the gut microbiota can have profound effects on the gut–microbiota–brain axis.

As previously mentioned, some neurological disorders are associated with gut problems, probably related to the modified microbiota due to immune environment change of the mucosa. Consequently, the restoration of mucosal immune homeostasis and the introduction of SCFA can indirectly modify the microbiota and, in turn, the gut–microbiota–brain axis, through the inhibition of histone deacetylase, producing benefit in the treatment of diseases such as Huntington’s disease, Parkinson’s disease and amyotrophic lateral sclerosis [137].

In addition, omega-3 polyunsaturated fatty acids (n-3-PUFA) may improve or prevent some neurological and neuroimmune disorders such as anxiety and/or depression [138]. Jiang et al. reported that serotonergic and dopaminergic neurotransmission was improved by docosahexaenoic acid (DHA) supplementation. DHA reduces also the levels of various adrenal–hypothalamus–pituitary axis hormones in mice, suggesting that DHA may be effective in the treatment of depression [138] and/or Alzheimer’s disease [139,140,141,142].

A randomized study about the n-3-PUFA supplements effect on the human gut microbiota showed an increase in SCFA-producing bacteria, such as Bifidobacterium, Roseburia, and Lactobacillus [143]. The same effect was also found in rats [144]. Thus, these results suggest that n-3-PUFA supplementation improves gastrointestinal homeostasis.

In addition, the gut microbiota composition of stressed rats could be altered by EPA and DHA supplementation [145], which, as is already known, are also present in fish such as salmon, mackerel, tuna, herring, and sardines.

Classified based on their structure as flavonoids or non-flavonoids, the polyphenols are largely present in many plants and fruits. More studies focused on the protective effects against cardiovascular disease, diabetes, cancer, dementia, as well as anti-aging effect [146,147,148,149].

Polyphenols influence the gut microbiota. Ankolekar et al. showed that gallic acid, quercetin, and tea catechins inhibit H. pylori [150]. Wang et al., after treating mice infected with E. coli O157: H7 with Fuzhuan brick tea extract, found improved immune function and increased microbiota diversity [151]. However, Janssens et al. found that long-term green tea supplementation does not change the human gut microbiota profile [152]. Among the food models proposed, the Mediterranean diet is recommended to improve health, quality, and even life expectancy, associating itself with a lower risk of diseases, including neurodegenerative diseases, such as Alzheimer’s disease, and Parkinson’s diseases, obesity, and cardiovascular diseases.

The greater constituents of this diet are vegetables, fruits, nuts, legumes, grains, fish and monounsaturated fatty acids, such as olive oil, while meat and dairy products are provided to a lesser extent. Furthermore, many bioactive nutrients, including polyphenols, retinoids, isothiocyanates, and some allyl compounds, molecules that intervene in the mechanisms of DNA repair, cell growth, and differentiation, mechanisms of apoptosis, oxidative stress, and inflammation, have been identified in the Mediterranean diet.

The healthy nutrition acting on gut microbiota and also on adipose tissue is involved in neurodegenerative disease development and/or prognosis (Figure 2).

Healthy nutrition and an adequate lifestyle act on the intestinal microbiota. The intestinal microbiota in turn influences the effects of nutrition, as well as the evolution and prognosis of neurodegenerative diseases, in a brain context influenced by adipokines produced by adipose tissue (Figure 2).

## 8. Conclusions

Until recently, it was thought that neuropsychiatric dysfunctions, mood, memory and cognitive process were regulated exclusively by the CNS through a fine regulation of hormonal factors. Today we have important scientific evidence that shows how the CNS is in constant communication with the adipose tissue, immune system and above all the intestinal microbiota, which influence each other. Neuro-immune activities at the CNS level are influenced directly through microbial metabolites or indirectly through systemic signals derived from the microbiota.

From this, it is clear that the human being is not only influenced by the environment in which he lives but also by the microorganisms that live in it and that have evolved with it. Therefore, preserving biodiversity within the microbial ecosystem, in terms of components and functions through an adequate diet, is of fundamental importance for promoting both physical and mental health conditions, preventing and/or treating some neurodegenerative diseases. The importance of correct nutrition is reflected also in its ability to act not only on gut microbiota but also on many inflammatory and immune mediators, among these, on Adiponectin. Furthermore, it is known that there is a strong interplay between Adiponectin, gut microbiota, and the brain–gut axis. For these reasons, correct nutrition can prevent many metabolic and inflammatory diseases such as neurodegenerative diseases and obesity, using diet components such as therapeutic molecules, which are able to prevent and/or treat neurodegenerative diseases. 

## Figures and Tables

**Figure 1 ijms-21-09255-f001:**
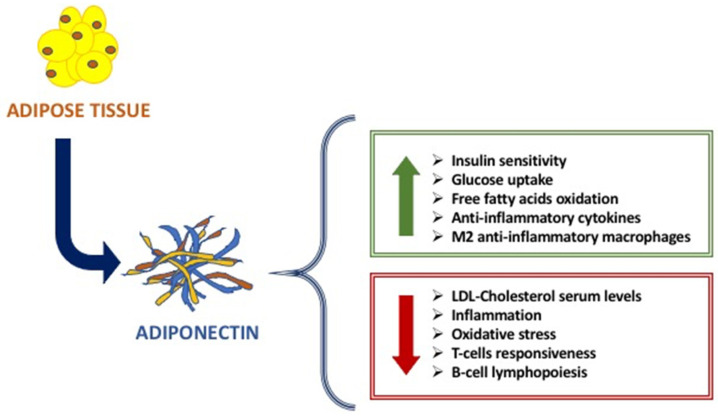
The main beneficial effects of Adiponectin.

**Figure 2 ijms-21-09255-f002:**
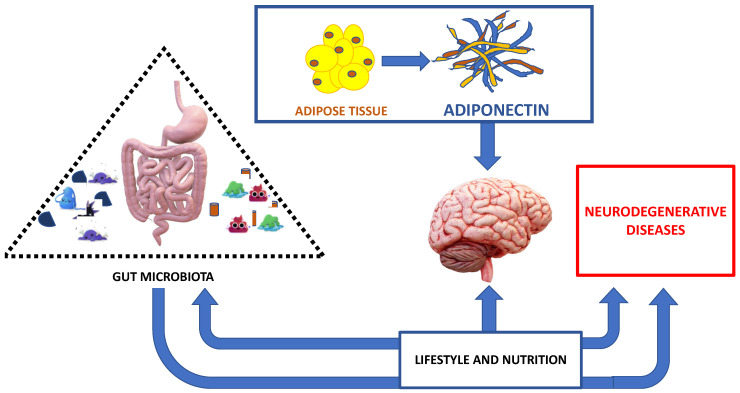
The major interplay between correct nutrition, adipose tissue, gut microbiota and neurodegenerative diseases.

**Figure 3 ijms-21-09255-f003:**
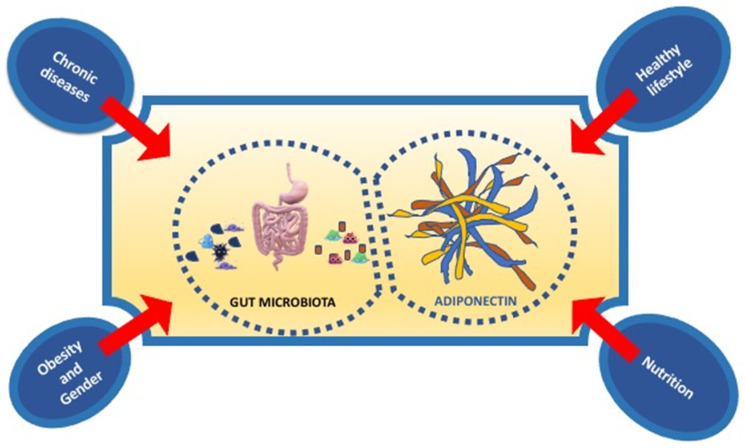
The principal factors influencing the interplay between gut microbiota and adiponectin.

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
