# Peer review of "Adiponectin Role in Neurodegenerative Diseases: Focus on Nutrition Review"

_ijms, 2020, doi:10.3390/ijms21239255_

Round 1

Reviewer 1 Report

This review article summarizes the role of Adiponectin in neurodegenerative diseases with a focus on nutrition. The manuscript is easy to read with timely information on neurodegenerative diseases, including Parkinson’s disease and Alzheimer’s disease, the gut microbiota, and the role of adiponectin in immune function. These authors also lay out the likely involvement of Adiponectin in the progress/regulation of the above-mentioned neurodegenerative diseases and how nutrients may interplay with gut microbiota to regulate these diseases. It should be a good entry point for readers seeking general information regarding the function of Adiponectin.

Following are some minor comments that need to be addressed prior to be published.

  1. Word choice. In the abstract and the context, these authors used “correct” diet, lifestyle or nutrition, which may be misleading because diet diversity and food preferences within a given population. Please revise.
  2. Authors need to provide references for the sentences began with “Many data” (page2, line49), “it is known” (page2, line54), “Studies In vitro” (page2 line 79), “More studies” (page3 line103). “Published data” (page3 line104), “Indeed, some authors” (page 7 line328).

Author Response

REVIEWER 1

This review article summarizes the role of Adiponectin in neurodegenerative diseases with a focus on nutrition. The manuscript is easy to read with timely information on neurodegenerative diseases, including Parkinson’s disease and Alzheimer’s disease, the gut microbiota, and the role of adiponectin in immune function. These authors also lay out the likely involvement of Adiponectin in the progress/regulation of the above-mentioned neurodegenerative diseases and how nutrients may interplay with gut microbiota to regulate these diseases. It should be a good entry point for readers seeking general information regarding the function of Adiponectin.

Following are some minor comments that need to be addressed prior to be published.

  1. Word choice. In the abstract and the context, these authors used “correct” diet, lifestyle or nutrition, which may be misleading because diet diversity and food preferences within a given population. Please revise.

We thank the reviewer for this suggestion, and in the abstract and in the text, we modified “correct” diet, lifestyle and nutrition with “healthy” diet, lifestyle and nutrition.

  1. Authors need to provide references for the sentences began with “Many data” (page 2, line 48), “it is known” (page 2, line 55), “Studies In vitro” (page 2 line 78), “More studies” (page 3 line104). “Published data” (page 3 line106), “Indeed, some authors” (page 9 line375).

We thank the reviewer for these suggestions. Therefore, we inserted the references required.

Reviewer 2 Report

In this review article the authors gives an overview of adiponectin role in neurodegenerative diseases with focus on nutrition, especially on “the gut-microbiota-brain axis”.

  1. Many sentences are included in the review without referring to the litterature. Please check the use of references throughout the manuscript. Provide references especially while referring to “previously studies have shown”. Please provide references in the introduction, where there are no references at all.

For example;

Page 2, line 76 “Studies In vitro and in vivo have highlighted the anti-inflammatory and anti-atherogenic effects of 76 adiponectin, linked to its action both suppressing the production of pro-inflammatory cytokines and 77 modulating of the expression of the anti-inflammatory cytokine IL-10 on various cell types such as 78 monocytes and macrophages ”

Page 3, line 101

“More studies have shown a different role of adiponectin in immune and autoimmune diseases. 101 In the literature it is reported that adiponectin is closely associated with immune diseases, both 102 immunodeficiencies and autoimmune diseases. Published data reported that serum adiponectin 103 levels increase in autoimmune diseases while decreasing in immunodeficiency diseases.”

  1. Page 3, line 99

Adiponectin concentrations are reduced in obesity. These lower levels of adiponectin in obese patients are related to several metabolic changes [7].”

Please provide the biological sample (serum, plasma etc.) where adiponectin concentrations are reduced. What kinds of metabolic changes are related to the adiponectin concentrations in the obese patients?

  1. Page 3, line 115

“Indeed, in rheumatoid arthritis, high levels of 116 Adiponectin are expressed in huge quantities and related to the disease progression [12]; it seems 117 that Adiponectin exerts pro-inflammatory actions, inducing pro-inflammatory cytokines production, 118 and activating Nf-kβ pathway.”

Expressed at mRNA level or protein level?

  1. Please check the sentence (Page 3, 105-108)

“The causes of this different modulation of Adiponectin in immune and autoimmune diseases 105 and the different anti or pro-inflammatory actions are not yet totally clear [8]; the explanation is that, 106 in addition to the different types of immune cells which intervene in immune diseases compared to 107 the autoimmune ones, different Adiponectin isoforms intervene.”

  1. Page 3, line 126 “In particular, these 126 diseases are characterized by extreme variability but, nevertheless, neuroinflammation and increased 127 intestinal permeability are common features [14].”

It seems that reference 14 is a review article. Please refer to the original articles.6

  1. Please describe the peripheral infections causing AD?

Page 4, line 175

“Some authors believe that AD pathogenesis is associated with a peripheral infectious origin, like to 175 herpes simplex virus type 1 (HSV1) infection (in mice), causing neuroinflammation [28-29].”

  1. Page 4, line 175-177,

“Some authors believe that AD pathogenesis is associated with a peripheral infectious origin, like to 175 herpes simplex virus type 1 (HSV1) infection (in mice), causing neuroinflammation [28-29]. Also, the 176 infections with Spirochaete, fungi, and Chlamydia pneumonia are considered potentially responsible 177 for AD [30-31]. Likewise, recent studies have implicated the gut microbiota in the etiology of AD.”

As far as I understand there is controversial that the microbiotics and the associations to the neurodegenerative diseases such as Alzheimer’s disease. (AD) Please provide more studies that support an association between AD and the microbiotic. Are there some consistent studies perform on human beings?

  1. Page 4, line 184

“In the post-mortem brain tissue of AD patients, as well as in vitro AD models, reduction in BDNF 184 has also been shown [35]. Due to recent evidence showing that BDNF levels are regulated by the gut 185 microbiota, it would be important to understand whether the altered microbiota can also contribute 186 to a reduction in BDNF in AD and thereby exacerbate the neuropathology of AD, oxidative stress, 187 and alter intestinal homeostasis in AD, as indeed it happens in the DP.”

Please provide the evidence how the BDNF levels are regulated by the microbiota? What is mean with DP?

  1. Section 4 “The gut-brain axis: the gut microbiota might be placed earlier in the manuscript.

Should the title of the section be changed to gut-microbiotica-brain axis instead?

Can it be useful to make an illustration of the gut-microbiota brain axis? Alternatively, move figure 1. It would possible be worthwhile spending more time explaining the vagus nerve (the parasympathic system and the sympathic system).

It would also be useful to provide an illustration on the intestinal lining/microbiota.

  1. Page 5, line 201

“In fact, the imbalance of the gut microbiota and/or a dysbiosis (increase in non-201 commensal microbes), determines a homeostasis alteration and disruption of the gut microbiota, also 202 altering the signaling between the gut-brain axis and producing neurological and neuroimmune 203 alterations. The dysbiosis consequences are the compromise of the intestinal epithelium and an 204 increased systemic inflammation with consequent up-regulation of IL-1, IL-6, and TNF-a [42].”

Is there neurodegenerative disorder investigated in reference 42?

  1. Please describe more general how inflammation influence the blood-brain-barrier.

  1. Page 6, line 267 “Scientific data shows the involvement of Adiponectin both in cerebrovascular disorders 267 and/or neurodegenerative diseases such as PD, AD, and in chronic inflammatory disease as multiple 268 sclerosis [80].

Provide references supporting evidence for involvement of adiponectin in PD and AD.

  1. Is there any studies that have measured adiponectin in CSF or brain tissue extract?

  1. Page 7, line 339

“The gut microbiota is an essential system to maintain physiological processes and health status.”

Please provide references.

  1. “Also, omega-3 polyunsaturated fatty acids (n-3-PUFA) may improve or prevent some 406 neurological and neuroimmune disorders.”

Which neurological and neuroimmune disorders?

Author Response

REVIEWER 2

In this review article the authors gives an overview of adiponectin role in neurodegenerative diseases with focus on nutrition, especially on “the gut-microbiota-brain axis”.

1. Many sentences are included in the review without referring to the litterature. Please check the use of references throughout the manuscript. Provide references especially while referring to “previously studies have shown”. Please provide references in the introduction, where there are no references at all.

We thank the reviewer and we checked and reported the lost references in the introduction section and in all text.

For example;

Page 2, line 81 “Studies in vitro and in vivo have highlighted the anti-inflammatory and anti-atherogenic effects of adiponectin, linked to its action both suppressing the production of pro-inflammatory cytokines and modulating of the expression of the anti-inflammatory cytokine IL-10 on various cell types such as monocytes and macrophages”

Page 3, line 106

“More studies have shown a different role of adiponectin in immune and autoimmune diseases. In the literature it is reported that adiponectin is closely associated with immune diseases, both immunodeficiency and autoimmune diseases. Published data reported that serum adiponectin levels increase in autoimmune diseases while decreasing in immunodeficiency diseases.”

2. Page 3, line 103

Adiponectin concentrations are reduced in serum of obese subject. These lower levels of adiponectin in obese patients are related to several metabolic changes, such as glucose uptake, lipid metabolism and insulin sensitivity [9].”

Please provide the biological sample (serum, plasma etc.) where adiponectin concentrations are reduced. What kinds of metabolic changes are related to the adiponectin concentrations in the obese patients?

 We corrected the text and reported such as:

“In particular, the metabolic, immune, and anti-inflammatory properties of adiponectin are evident in obesity. Adiponectin concentrations are reduced in the serum of obese subjects. These lower serum levels of adiponectin in obese patients are related to several metabolic changes, such as glucose uptake, lipid metabolism, and insulin sensitivity”

3. Page 3, line 115

“Indeed, in rheumatoid arthritis, high levels of 116 Adiponectin are expressed in huge quantities and related to the disease progression [12]; it seems 117 that Adiponectin exerts pro-inflammatory actions, inducing pro-inflammatory cytokines production, 118 and activating Nf-kβ pathway.”

 We thank the reviewer for this observation, and we specify in the text that we refer to the protein serum level. In the text we reported as:

“Indeed, in rheumatoid arthritis, high serum levels of Adiponectin are expressed in huge quantities and related to the disease progression; it seems that Adiponectin exerts pro-inflammatory actions, inducing pro-inflammatory cytokines production, and activating the pathway of Nf-kβ protein complex.”

4.Please check the sentence (Page 3, 105-108)

“The causes of this different modulation of Adiponectin in immune and autoimmune diseases 105 and the different anti or pro-inflammatory actions are not yet totally clear [8]; the explanation is that, 106 in addition to the different types of immune cells which intervene in immune diseases compared to 107 the autoimmune ones, different Adiponectin isoforms intervene.”

 We revised this sentence and reported as:

“The causes of this different modulation of Adiponectin in immunodeficiencies and autoimmune diseases and the different anti or pro-inflammatory actions of Adiponectin are not yet totally clear [8]; the explanation is that, in addition to the different types of immune cells and mechanism which intervene in immunodeficiencies compared to the autoimmune diseases, different Adiponectin isoforms intervene. Indeed, the low molecular weight adiponectin oligomers seems to induce an pro-inflammatory actions of adiponectin in autoimmune diseases, while high molecular weight adiponectin oligomers induces and anti-inflammatory actions of adiponectin, especially in immunodeficiencies”

  1. Page 3, line 126 “In particular, these 126 diseases are characterized by extreme variability but, nevertheless, neuroinflammation and increased 127 intestinal permeability are common features [14].”

It seems that reference 14 is a review article. Please refer to the original.

We checked this article, and we added "Review" in the References section at 20 position

  1. Please describe the peripheral infections causing AD?

Page 4, line 175

“Some authors believe that AD pathogenesis is associated with a peripheral infectious origin, like to 175 herpes simplex virus type 1 (HSV1) infection (in mice), causing neuroinflammation [28-29].”

In the manuscript we reported  (page 5 line 187) as:

“Some authors believe that AD pathogenesis is associated with a peripheral infectious origin, like to herpes simplex virus type 1 (HSV1) infection (in mice), causing neuroinflammation. Also, the infections with Spirochaete, fungi, and Chlamydia pneumonia are considered potentially responsible for AD. Likewise, recent studies have implicated the gut microbiota in the etiology of AD”

  1. Page 4, line 175-177,

“Some authors believe that AD pathogenesis is associated with a peripheral infectious origin, like to 175 herpes simplex virus type 1 (HSV1) infection (in mice), causing neuroinflammation [28-29]. Also, the 176 infections with Spirochaete, fungi, and Chlamydia pneumonia are considered potentially responsible 177 for AD [30-31]. Likewise, recent studies have implicated the gut microbiota in the etiology of AD.”

As far as I understand there is controversial that the microbiotics and the associations to the neurodegenerative diseases such as Alzheimer’s disease. (AD) Please provide more studies that support an association between AD and the microbiotic. Are there some consistent studies perform on human beings?

We thank the reviewer and we explain (page 5 line 187-200) as :

“Some authors believe that AD pathogenesis is associated with a peripheral infectious origin, like to herpes simplex virus type 1 (HSV1) infection (in mice), causing neuroinflammation. Also, the infections with Spirochaete, fungi, and Chlamydia pneumonia are considered potentially responsible for AD. Likewise, recent studies have implicated the gut microbiota in the etiology of AD. Indeed, as reported by Miklossy the infectious agents can initiate the degenerative process in AD, sustain chronic inflammation, and lead to progressive neuronal damage and amyloid    deposition.    The    accumulated knowledge, views and hypotheses proposed to explain the pathogenesis of AD fit well with an infectious origin of the disease. The outcome of infection   is   determined   by   the   genetic predisposition of the patient, by the virulence and biology of the infecting agent, and by various environmental factors, such as exercise,stress and nutrition. Furthermore, there is a significant association between Alzheimer disease (AD) and various types of spirochete and other pathogens such as Chlamydophyla pneumoniae and herpes simplex virus type-1 (HSV-1). Exposure of mammalian neuronal and glial cells and organotypic cultures to spirochetes reproduces the biological and pathological hallmarks of AD.”

8. Page 4, line 184

“In the post-mortem brain tissue of AD patients, as well as in vitro AD models, reduction in BDNF 184 has also been shown [35]. Due to recent evidence showing that BDNF levels are regulated by the gut 185 microbiota, it would be important to understand whether the altered microbiota can also contribute 186 to a reduction in BDNF in AD and thereby exacerbate the neuropathology of AD, oxidative stress, 187 and alter intestinal homeostasis in AD, as indeed it happens in the DP.”

Please provide the evidence how the BDNF levels are regulated by the microbiota? What is mean with DP?

 We thank the reviewer, and we added in the text (page  5 line 210-213):

Indeed, as shown in a murine model, by Chen et al, BDNF regulates colonic mucosal barrier function and affect gut microbiota where the expression of ZO-1, occludin, claudin-1 and claudin-2 plays an important role in maintaining the intestinal mucosal function”.

PD means Parkinson 'Disease, as specified on page 4 line 144.

9. Section 4 “The gut-brain axis: the gut microbiota might be placed earlier in the manuscript.

Should the title of the section be changed to gut-microbiotica-brain axis instead?

We thank the reviewer for this suggestion. We corrected the title as you requested. Regarding the section, we think that is ok in this order because the focus of our review is on adiponectin in neurodegenerative diseases.

Can it be useful to make an illustration of the gut-microbiota brain axis? Alternatively, move figure 1. It would possible be worthwhile spending more time explaining the vagus nerve (the parasympathic system and the sympathic system).

It would also be useful to provide an illustration on the intestinal lining/microbiota.

We added in this section:

The vagus nerve (VN) is a component of the parasympathetic nervous system that actively participates to homeostasis and interactions between gut microbiota and brain. Is known that nerve disturbances are the evidence of central nervous system dysfunction, such as mood disturbances or neurodegenerative diseases, but also gastrointestinal disorders, such as irritable bowel syndrome. [46-50]. Previous studies have indicated that vagal efferent fibers regulate responses of the gastrointestinal system through the release of neurotransmitters [50]. Less activation of the vagal nerve causes excessive production and activation of neurotransmitters, thus compromising the digestive process and affecting gastric motility [47-49]. Furthermore, immune effects regulated by the vagal nerve were also observed. Studies have shown that M1 macrophage activation and increased proinflammatory cytokine

In addition, we moved the figure 1 in this section.

10. Page 5, line 201

“In fact, the imbalance of the gut microbiota and/or a dysbiosis (increase in non-201 commensal microbes), determines a homeostasis alteration and disruption of the gut microbiota, also 202 altering the signaling between the gut-brain axis and producing neurological and neuroimmune 203 alterations. The dysbiosis consequences are the compromise of the intestinal epithelium and an 204 increased systemic inflammation with consequent up-regulation of IL-1, IL-6, and TNF-a [42].”

Is there neurodegenerative disorder investigated in reference 42?

No, because in this section we described the dysbiosis. Next we reported it, referring specifically to neurodegenerative disorders.

“….When a systemic inflammatory state is established, the blood-brain barrier (BBB) is altered, favoring consequently a state of neuroinflammation. Biesmans et al demonstrated that a single intraperitoneal injection of TNF-α in mice increases serum and brain levels of pro-inflammatory cytokines (TNF-α, IL-6 and MCP-1)” (Page 6 line 241-245)

12. Page 6, line 267 “Scientific data shows the involvement of Adiponectin both in cerebrovascular disorders 267 and/or neurodegenerative diseases such as PD, AD, and in chronic inflammatory disease as multiple 268 sclerosis [80].

Provide references supporting evidence for involvement of adiponectin in PD and AD.

We added the references that reported the adiponectin involvement in PD and AD.

13. Is there any studies that have measured adiponectin in CSF or brain tissue extract?

In the text (page 7 line 311-314) we added:

“In addition, data literature reported that the Adipoenctin in cerebrospinal fluid inversely correlates to AD progression. Furthermore, Adiponectin was also colocalized with p-tau in neurofibrillary tangles in AD, suggesting that sequestering may occur, which could explain the reduced CSF adiponectin levels”

14. Page 7, line 339

“The gut microbiota is an essential system to maintain physiological processes and health status.”

Please provide references.

We added the references in the text (Ref 4,20)

15. “Also, omega-3 polyunsaturated fatty acids (n-3-PUFA) may improve or prevent some 406 neurological and neuroimmune disorders.”

Which neurological and neuroimmune disorders?

We reported in the text (page 11 line 461-462) as:

“Also, omega-3 polyunsaturated fatty acids (n-3-PUFA) may improve or prevent some neurological and neuroimmune disorders such as anxiety and/or depression.”

Reviewer 3 Report

The manuscript entitled: “Adiponectin role in neurodegenerative diseases: Focus on nutrition -review” presents an interesting overview regarding the interaction between the gut microbiota, adiponectin and brain-gut axis and how nutrition can modulate the development of neurodegenerative diseases and obesity. The manuscript is very interesting but some issues should be improved before acceptance.

  • In the introduction, you state a lot of facts without the support of references. Please revise and add references accordingly. You can check also: doi: 10.3389/fphys.2020.00694; https://doi.org/10.3390/ijms21175964; https://doi.org/10.3390/ijms21030786; DOI: 3390/medicina56090433; doi: 10.1016/j.toxlet.2019.04.014. 
  • In part 2 of the manuscript where you describe the role of adiponectin in the organism a figure that summarize the roles of adiponectin will be very useful for the reader.
  • The paragraph from line 101-104: “More studies have shown a different role of adiponectin in immune and autoimmune diseases. In the literature, it is reported that adiponectin is closely associated with immune diseases, both immunodeficiencies and autoimmune diseases. Published data reported that serum adiponectin levels increase in autoimmune diseases while decreasing in immunodeficiency diseases.” needs references to support the statement.
  • For the chapter: The interplay between Adiponectin and gut microbiota please provide a schematic flowchart of the up to know information regarding the interaction between adiponectin and gut microbiota.

Round 2

Reviewer 2 Report

-

Reviewer 3 Report

The authors addressed all the comments of the reviewer and the manuscript is much improved compared with the initial version. I suggest acceptance.